# Amphiphilic Diblock Copolymers Bearing Poly(Ethylene Glycol) Block: Hydrodynamic Properties in Organic Solvents and Water Micellar Dispersions, Effect of Hydrophobic Block Chemistry on Dispersion Stability and Cytotoxicity

**DOI:** 10.3390/polym14204361

**Published:** 2022-10-16

**Authors:** Anastasiia A. Elistratova, Alexander S. Gubarev, Alexey A. Lezov, Petr S. Vlasov, Anastasia I. Solomatina, Yu-Chan Liao, Pi-Tai Chou, Sergey P. Tunik, Pavel S. Chelushkin, Nikolai V. Tsvetkov

**Affiliations:** 1Institute of Chemistry, St. Petersburg State University, Universitetskii Av., 26, 198504 St. Petersburg, Russia; 2Department of Molecular Biophysics and Physics of Polymers, St. Petersburg State University, Universitetskaya nab., 7/9, 199034 St. Petersburg, Russia; 3Department of Chemistry, National Taiwan University, No. 1, Sec. 4, Roosevelt Rd., Taipei 10617, Taiwan

**Keywords:** block copolymer micelles, dynamic light scattering, analytical ultracentrifugation, cytotoxicity

## Abstract

Despite the fact that amphiphilic block copolymers have been studied in detail by various methods both in common solvents and aqueous dispersions, their hydrodynamic description is still incomplete. In this paper, we present a detailed hydrodynamic study of six commercial diblock copolymers featuring the same hydrophilic block (poly(ethylene glycol), PEG; degree of polymerization is ca. 110 ± 25) and the following hydrophobic blocks: polystyrene, PS_35_-*b*-PEG_115_; poly(methyl methacrylate), PMMA_55_-*b*-PEG_95_; poly(1,4-butadyene), PBd_90_-*b*-PEG_130_; polyethylene PE_40_-*b*-PEG_85_; poly(dimethylsiloxane), PDMS_15_-*b*-PEG_115_; and poly(ɛ-caprolactone), PCL_45_-*b*-PEG_115_. The hydrodynamic properties of block copolymers are investigated in both an organic solvent (tetrahydrofuran) and in water micellar dispersions by the combination of static/dynamic light scattering, viscometry, and analytical ultracentrifugation. All the micellar dispersions demonstrate bimodal particle distributions: small compact (hydrodynamic redii, R_h_ ≤ 17 nm) spherical particles ascribed to “conventional” core–shell polymer micelles and larger particles ascribed to micellar clusters. Hydrodynamic invariants are (2.4 ± 0.4) × 10^−10^ g cm^2^ s^−2^ K^−1^ mol^−1/3^ for all types of micelles used in the study. For aqueous micellar dispersions, in view of their potential biomedical applications, their critical micelle concentration values and cytotoxicities are also reported. The investigated micelles are stable towards precipitation, possess low critical micelle concentration values (with the exception of PDMS_15_-*b*-PEG_115_), and demonstrate low toxicity towards Chinese Hamster Ovarian (CHO-K1) cells.

## 1. Introduction

The fundamental property of amphiphilic block copolymers is their ability to undergo nanophase separation in solvents selective towards at least one of the blocks [1,2]. In the case of water, this process results in formation of nanosized aggregates (usually referred to as block copolymer micelles) featuring core-shell topology: the inner core is formed by contracted hydrophobic blocks while the outer shell (corona) consists of swollen hydrophilic chains [1,2]. Morphology and sizes of such block copolymer micelles vary strongly depending on the block copolymer chemistry and preparation method, but the most common structures are spherical sub-50 nm particles typically formed by A-B or A-B-A di- or triblock copolymers with degrees of polymerization (DP) ranging from 50 to 500 for hydrophilic (A), and from 20 to 200 for hydrophobic (B) blocks, respectively, with DP(A)/DP(B) > 1. The most typical block chemistries described in the literature are hydrophilic poly(ethylene glycol), PEG, and hydrophobic polystyrene, PS, or poly(ɛ-caprolactone), PCL.

Due to the beneficial combination of practically important properties (small size, high dispersion stability, and ability to solubilize hydrophobic compounds), aqueous dispersions of block copolymer micelles find numerous practical applications as nanocontainers for drugs [3,4], imaging and theranostic agents [5,6,7,8,9,10], nanoreactors for micellar catalysis [11] or nanoparticle synthesis [12,13,14], etc. Almost all of the aforementioned applications require an as narrow as possible size distribution of the nanoparticles. Unfortunately, polymer micelles based on nonionic amphiphilic block copolymers with poly(ethylene glycol) PEG coronas are often reported to form non-negligible amounts of larger aggregates in aqueous dispersions [15,16,17]. The formation of water dispersions of block copolymer micelles lacking secondary association is still a subject of controversy: depending on numerous parameters (block copolymer synthesis history and its dispersity, details of micelle preparation protocol, etc.), different reports describe either the absence [18,19] or presence [15,16,17] of secondary aggregates in micellar dispersions for almost identical starting block copolymers. In practice, each newly synthesized block copolymer intended for application in micellar form should be first evaluated for its ability to form non-aggregated micelles.

Further, despite the fact that nonionic amphiphilic diblock copolymers were studied in detail by various methods (predominantly, structural investigation in aqueous dispersions by small-angle X-ray [18,20] and neutron [21] scattering, dynamic light scattering [15,16,18,21], and morphological studies by electron microscopy [22,23]), their hydrodynamic description is still incomplete. At present, systematic studies are available only for several block copolymers, such as PS-*b*-PEG [1,13,16,17,18,20,21,22,23,24,25] and PCL-*b*-PEG [10,26,27,28,29,30,31], while other promising systems such as poly(methyl methacrylate) (PMMA-*b*-PEG [32]) or poly(butadiene) (PBd-*b*-PEG [12]) are described in sporadic publications. Moreover, a typical paradigm used in these studies (i.e., restricting to some particular block copolymer chemistry and focusing on block lengths variations) does not allow a comparison of different block chemistries within the same experimental methodology.

In this context, we present the results of our investigation of six commercial amphiphilic block copolymers with poly(ethylene glycol) hydrophilic blocks of comparable DP and different hydrophobic blocks (different in chemistry, but also comparable in their DP). The commercial block copolymer samples were chosen because of their broad availability. We investigate the ability of these block copolymers to form stable micellar aqueous dispersions, describe in detail their hydrodynamic properties in both organic solvents and micellar aqueous dispersions by the combination of static/dynamic light scattering (SLS/DLS), viscometry, and analytical ultracentrifugation (AUC), with a special attention to secondary micelle aggregation. Additionally, for micellar aqueous dispersions, in view of their potential biomedical applications, we assess the cytotoxicity of the corresponding block copolymer micelles. To the best of our knowledge, a comparison within the uniform methodology (combination of SLS/DLS and AUC supported by gel permeation chromatography, viscometry, gel permeation chromatography, GPC, and ^1^H NMR) of a set of block copolymers with close block lengths and the same hydrophilic PEG block but rather different hydrophobic block chemistries has not been reported to date.

## 2. Materials and Methods

### 2.1. General Comments

PS_35_-*b*-PEG_115_ (Sample #: P10086B-SEO), PMMA_55_-*b*-PEG_95_ (Sample #: P5164-EOMMA), PBd_90_-*b*-PEG_130_ (Sample #: P4603-BdEO), PE_40_-*b*-PEG_85_ (Sample #: P3288-EEO), and PDMS_15_-*b*-PEG_115_ (Sample #: P7261-DMSEO) were purchased from Polymer Source, Dorval, QC, Canada; PCL_45_-*b*-PEG_115_ was purchased from Sigma-Aldrich, St Louis, MO, USA (Product number: 570303). Ratios of DP(PEG)/DP(hydrophobic block) for all the block copolymers (except for PE-*b*-PEG) were calculated from ^1^H NMR spectra (Bruker Avance 400 spectrometer; Bruker, Ettlingen, Germany); all spectra were measured in DMSO-*d_6_* at room temperature.

*N*,*N*-dimethylformamide (DMF; Ekos-1, Moscow, Russia) and dimethylsulfoxide (DMSO; Vecton, Saint Petersburg, Russia) were distilled in vacuum using standard procedures. Tetrahydrofuran (THF) and 1,4-dioxane were purified by distillation over sodium hydroxide in argon atmosphere. Acetonitrile (chromatographic “0” grade; KryoChrom, Saint Petersburg, Russia) and *N*-methylpyrrolidone (NMP; Sigma-Aldrich, St Louis, MO, USA) were used as received. Water was purified using a Simplicity water purification system Merck Millipore, San Jose, CA, USA (type 1 water).

### 2.2. Preparation of Block Copolymer Micelles

The general preparation protocol for block copolymer micelles consisted in dissolving block copolymers in organic solvent (DMF, DMSO, THF, 1,4-dioxane, NMP) and slight heating to 40 °C to yield transparent solutions. Then, 3 volumes of type 1 water were added dropwise to the solutions under intensive stirring (1200 rpm) to form the micelle dispersions followed by removal of organic co-solvents by dialysis using dialysis tubes with a molecular weight cut-off of 14 kDa (Sigma-Aldrich, St Louis, MO, USA) against type 1 water (5–7 water changes). The obtained micelle dispersions were stored in vials in a fridge in dark at 4 °C. Block copolymer concentrations were confirmed by weighting the residues after freeze-drying of pre-weighted aliquots of micellar dispersions. UV/Vis spectra of micellar dispersions were recorded using a UV-1800 spectrophotometer (Shimadzu, Kyoto, Japan) in water in 10 mm absorption quartz glass cells (Hellma Analytics, Müllheim, Germany).

### 2.3. Gel Permeation Chromatography (GPC)

GPC was performed on Prominence LC-20AD (Shimadzu, Kyoto, Japan) chromatograph equipped with a refractometric detector and PLgel MIXED-C column (300 × 7.5 mm, 5 µm particles, linear molecular weight range up to 2000 kg/mol based on polystyrene, Agilent Technologies, Amstelveen, The Netherlands). Runs were performed in THF at 40 °C and 1.0 mL/min flow rate, P = 4.2–4.3 MPa. Block copolymer solutions (3 mg/mL) were filtered through 0.22 µm PTFE filters. Weight average molar masses (*M_w_*) and dispersities (*Đ* = *M_w_*/*M_n_*) were calculated from GPC traces (Appendix A) using the LCSolution software (Version 1.22; Shimadzu, Kyoto, Japan) and summarized in Table 1. Cubic calibration curve was built using a set of polystyrene standards (500–250,000 g/mol).

### 2.4. Dynamic Light Scattering (DLS)

DLS experiments were performed on block copolymer solutions in THF or micellar dispersions in water. The samples were placed into cylindrical cuvettes and centrifuged for 30 min at 5000 rpm before investigation.

The PhotoCor-Complex (PhotoCor Instruments Inc., Moscow, Russia) experimental setup was used. It was equipped with a real-time correlator (288 channels, minimal t=20 ns); laser (λ=654 nm) was used as an excitation source; the experiments were carried out at scattering angles (ϑ) ranging from 30° to 140° at a temperature of 20 ± 0.1 and 25 ± 0.1 °C for THF and H_2_O solutions, correspondingly. Autocorrelation functions of scattered light intensity $G^{\left(2\right)}\left(t\right)=\langle I\left(t_{0}\right)I\left(t_{0}+t\right)\rangle /\langle I{\left(t_{0}\right)\rangle }^{2}$ were processed using DynaLS software (Version 2.7.1.; PhotoCor Instruments Inc., Moscow, Russia). It provides distributions I(τ) of scattered light intensities by relaxation times *τ* in accordance with the relation: $G^{\left(1\right)}\left(t\right)=\int^{\;}E\left(\tau \right){\;e}^{-t/\tau }d\tau$, where $G^{\left(1\right)}\left(t\right)$ is related to $G^{\left(2\right)}\left(t\right)$ by Siegert relation $G^{\left(2\right)}\left(t\right)=B+\beta {\left|G^{\left(1\right)}\left(t\right)\right|}^{2}$, here B is base line, and β is coherence factor.

Translational diffusion coefficients *D* at fixed concentrations were calculated from the slope of this line according to the following relationship: $1/\tau =Dq^{2}$, where *q* is the wave vector. The diffusion coefficients $D_{0}$ were determined by extrapolation of D(c) dependence to infinite dilution according to equation: $D\left(c\right)=D_{0}\left(1+c2A_{2}M\right)$, where *A_2_* is the second virial coefficient.

Hydrodynamic radii *R*_h_ were calculated using the Stokes–Einstein equation:(1)$$R_{h}=k_{B}T/\left(6\pi \eta_{0}D_{0}\right)$$
where *k*_B_ is the Boltzmann constant, *T* the temperature, and *η*_0_ the shear viscosity of the solvent.

The mass fractions c of micelles were estimated using the following relationship:(2)$$c_{i}=\omega_{i}/R_{hi}^{\alpha }$$
where $R_{hi}$ is hydrodynamic radius of the *i*-th particle, $\omega_{i}$ its contribution to the scattered light intensity. The exponent α depends on the shape of the particles, and for the spherical ones α=3.

Weight-average molar masses *M_w_* of the copolymer micelles were determined from the static light scattering data according to the following equation:(3)$${\frac{Hc}{R_{\theta }}|}_{\theta \to 0}=\frac{1}{M_{w}}+2A_{2}c,$$
where $H=4\pi^{2}n^{2}{\left(\frac{\partial n}{\partial c}\right)}^{2}/\left(\lambda^{4}N_{A}\right),$
*R_θ_* is the Rayleigh ratio, *A*_2_ is the second virial coefficient, ∂n/∂c is the refractive index increment, and *N_A_* is the Avogadro number. The Rayleigh ratio was calculated from the equation $R_{\theta }={\left(\frac{n_{0}}{n_{T}}\right)}^{2}\frac{I_{s}\left(\theta \right)-I_{0}\left(\theta \right)}{I_{T}\left(\theta \right)}\left(R_{T}I_{p}\left(\theta \right)\right)$, where *I_S_*(*θ*), *I*_0_(*θ*), *I_T_*(*θ*) are scattered light intensities of the studied solution, solvent and toluene at a fixed angle *θ*, *R_T_* is toluene Rayleigh ratio, *n_T_* is toluene refractive index, *I_p_*(*θ*) is an area of the peak on *I*(*τ*) distribution that corresponds to individual micelles.

### 2.5. Analytical Ultracentrifugation (AUC)

Sedimentation velocity experiments were performed with a ProteomeLab XLI Protein Characterization System analytical ultracentrifuge (Beckman Coulter, Brea, CA, USA), using conventional 12 mm aluminum (THF solutions) and Epon-Charcoal (H_2_O, D_2_O/H_2_O) double-sector centerpieces and a four-hole rotor (An-60Ti). Rotor speed range was 15,000–60,000 rpm, depending on the studied systems. The maximum possible rotor speeds were used for studying starting copolymer samples in THF solutions. The cells were filled with 420 μL of a sample solution and corresponding solvent in the reference sector. Before the run, the rotor was thermostated for approximately 2 h at 20 °C in the centrifuge vacuumed chamber for THF solutions (25 °C for water solutions). Sedimentation profiles were predominantly obtained with interference optical system at the same temperature.

The analysis of the sedimentation velocity data was performed using *c(s)* model with a Tikhonov–Phillips regularization procedure implemented into the Sedfit software, version 16.1c [33]. The *c(s)* analysis is based on the numerical solution of the Lamm equation assuming the averaging frictional ratio (*f*/*f_sph_*) values for all sedimenting species. The solvent hydrodynamic properties were determined experimentally (Appendix A).

While calculating M*_sD_*, the precise value of partial specific volume $\bar{\upsilon }$ is necessary. Whenever it is possible, it can be determined through standard protocols of densitometry measurements; however, when the highest possible concentration of copolymer micelles is limited and its value is imprecise, then an alternative technique (known as the *density variation approach* [34,35]) can be implemented. It requires isotopically different solvents that differ in basic solvent parameters (i.e., density and dynamic viscosity), and at the same time the conformational status of macromolecules is not affected. In our case, such a solvent can be D_2_O. Then, the comparison of sedimentation coefficients, obtained in the H_2_O and D_2_O/H_2_O mixture, should give the genuine value of partial specific volume:(4)$$\bar{\upsilon }=\frac{s_{H2O}\eta_{0H2O}-s_{D2O}\eta_{0D2O}}{s_{H2O}\eta_{0H2O}\rho_{0D2O}-s_{D2O}\eta_{0D2O}\rho_{0H2O}}$$

This approach also has some limits of application, as was shown elsewhere [36], as the solutions of salts in the regular and deuterated solvents reveal different structures. The accuracy of determining the partial specific volume in this manner is also negatively affected by dispersity of the studied polymers. Partial specific volumes of copolymer micelles determined by the aforementioned approach are presented in Appendix A. It can be seen that there is a significant difference between the values of starting block copolymers and the corresponding micelles based on them, which implies that the partial specific volume of micelles in water systems should be carefully determined.

### 2.6. Viscometry

The intrinsic viscosities of the block copolymer solutions and micellar dispersions were obtained by a Lovis 2000 M rolling-ball micro viscometer (Anton Paar, Graz, Austria) at 20 and 25 °C for THF and H_2_O solutions, correspondingly. The values of dynamic viscosity of a solution *η* were averaged over a series of angles and used in calculations of specific viscosity $\eta_{sp}/c=\left(\eta -\eta_{0}\right)/\eta_{0}c$, where *c* is the concentration of a solution, $\eta_{0}$—solvent viscosity. The values of intrinsic viscosity of studied solutions were calculated from extrapolation of dependences of $\eta_{sp}/c\left(c\right)$ to infinite dilution according the equation $\left[\eta \right]=\underset{c\to 0}{\lim}\frac{\eta_{sp}}{c}$. The slope of $\eta_{sp}/c\left(c\right)$ according equation $\frac{\eta_{sp}}{c}=\left[\eta \right]+k_{H}{\left[\eta \right]}^{2}c$ is defined by the dimensionless Huggins constant $k_{H}$, which depends on the thermodynamic solvent quality (for *θ*-solvents, *k_H_* = 0.4–0.7; for good solvents, *k_H_* = 0.2–0.4 [37,38]).

Alternatively, [*η*] can be obtained by linear extrapolation of $\frac{ln\eta_{r}}{c}$ vs. *c* to infinite dilution according to the Kraemer equation: $\frac{ln\eta_{r}}{c}=\left[\eta \right]-k_{K}{\left[\eta \right]}^{2}$. Here, $\eta_{r}=\eta_{0}$ and *k_K_* is Kraemer constant. The value of *k_K_* related to the Huggins constant as $k_{H}+k_{K}=0.5$, but it should be noted that the validity of this relationship follows from purely mathematical assumptions.

### 2.7. Cell Culture and Cell Viability (MTT) Assay

Chinese hamster ovary (CHO-K1) cell line was obtained from the Russian Cell Culture Collection (Institute of Cytology, St. Petersburg, Russia). Cell culture maintaining was performed as described in our recent publication [39]. The toxicity of block copolymer micelles in vitro was evaluated using an MTT assay [40]. The CHO-K1 cells were seeded in a 96-well plate (Nunc, Thermo Fisher Scientific, Waltham, MA, USA) at a concentration of 10^4^ cells per well and allowed to adhere for 24 h at 37 °C, 5% CO_2_ in standard culture media (Dulbecco’s Modified Eagle Medium/Nutrient Mixture F-12, DMEM/F12; 10% Fetal bovine serum, FBS; 100 Uints penicillin/streptomycin). The dispersions of block copolymer micelles in water were added to the cells at concentrations of 0.05, 0.1, 0.15, 0.2, 0.25, and 0.3 mg/mL. For each concentration, 12 replications were completed. After 24 h of incubation with the samples, the medium was removed from each well and replaced with 100 µL of fresh culture medium containing 0.25 mg/mL of 3(4,5-dimethyl-2-thiasolyl)-2,5-diphenyl-2H-tetrasole bromide (MTT; Thermo Fisher Scientific, Waltham, MA, USA). After 2–3 h of incubation, the MTT-containing media was replaced with 100 µL of DMSO (biology grade, Helicon, Moscow, Russia) in each well and incubated for 15 min to dissolve the formazan crystals. The plates were shacked thoroughly using an orbit plate shaker, and then the absorbance at 570 nm was measured by using a SPECTROstar Nano microplate reader (BMG LABTECH, Ortenberg, Germany). The viability of cells was calculated as the ratio of the sample’s optical density to that of the control.

## 3. Results and Discussion

The first part of the present study deals with the investigation of hydrodynamic behavior of a series of amphiphilic block copolymers in organic solvent (tetrahydrofuran). The second part describes the preparation of block copolymer micelles in water, assessment of their stability (towards preparative centrifugation, freeze-drying, and dilution), and hydrodynamic investigation by means of dynamic (DLS) and static (SLS) light scattering and analytical ultracentrifugation (AUC). The final part of the study explores cytotoxicity of the obtained micellar dispersions.

### 3.1. Block Copolymers Used in the Study

We used a series of six commercially available nonionic diblock copolymers with the same poly(ethylene glycol) (PEG) block and different hydrophobic blocks (Figure 1): polystyrene (PS-*b*-PEG), poly(methyl methacrylate) (PMMA-*b*-PEG), polybutadiene (rich in 1,4- microstructure, PBd-*b*-PEG), polyethylene (PE-*b*-PEG), poly(dimethyl siloxane) (PDMS-*b*-PEG), and polycaprolactone (PCL-*b*-PEG). In Figure 1, DP values listed in the subscripts of block copolymer abbreviations were calculated using the *M_n_* values provided by the manufacturers even in the case of more or less substantial differences with the obtained experimental data (Table 1); in the text, due to rather similar DP for PEG and hydrophobic blocks, the DP values are omitted for simplicity. It is important to note that, according to preparation schemes provided by the manufacturer (see Part 1 of Appendix A for more detail), at least four block copolymers (PS-*b*-PEG, PBd-*b*-PEG, PE-*b*-PEG, and PDMS-*b*-PEG) can potentially contain hydrophobic homopolymer (unreacted macroinitiator) as an admixture. This possibility should be kept in mind in the discussion of hydrodynamic behavior of block copolymers in organic solvents and the secondary aggregation of block copolymer micelles in aqueous dispersion.

By choosing the types of hydrophobic blocks (comprising micellar cores), we tried to bring together and compare (within the uniform experimental methodology) the chemistries, which are as different as possible in the following terms:Glass transition (*T_g_*) temperature; this parameter governs the ability of micelles to equilibrate; the micelles with “glassy” cores (i.e., composed of blocks with *T_g_* > room temperature, RT) are assumed to be irreversible (“frozen”) ones and *vice versa* [2]. In turn, ‘frozen’ micelles are much more stable towards any rearrangements upon variations of ambient conditions and disintegration upon dilution, and this feature is quite beneficial in numerous applications. In the set of polymers we used, PS-*b*-PEG and PMMA-*b*-PEG (with *T_g_* are of ca. 100 °C in bulk [41]) are expected to form “frozen” micelles while PDMS-*b*-PEG, PE-*b*-PEG, and PBd-*b*-PEG (bulk *T_g_* for PDMS, PE, and PBd are of ca. −120 °C, −80… −120 °C, and −60… −100 °C, respectively [41]), are expected to display much higher chain mobility inside cores; in the case of PCL-*b*-PEG, the core structure is more complex, since bulk PCL is a semicrystalline polymer with low *T_g_* of −60 °C, but a high melting point of 60 °C [42];Hydrophobicity; the higher hydrophobicity, the lower critical micelle concentration (CMC); additionally, micellar cores composed of highly hydrophobic blocks (such as PS) were reported to be almost free of water. In our set, most block copolymers are strongly hydrophobic; nevertheless, PCL and PMMA blocks contain relatively polar ester groups and can potentially be plasticized by water to some extent;Gas permeability; this requirement is not general and relates to our recent study where we have outlined the prospects of polymer micelles application in intracellular lifetime oxygen biosensing [39,43]: in this particular aspect, block copolymer micelles serve as nanocontainers for phosphorescent organometallic complexes that rapidly and reversibly respond to the changes in oxygen concentration by varying their luminescence lifetime. We have shown that, in the micelles, the hydrophobic phosphors are embedded into micellar cores, where the outer shell strongly protects the reporter molecule from interactions with biomolecules, thereby preserving its lifetime response from various biasing factors [39]. Obviously, the highest oxygen sensing response can be anticipated in the case of high gas permeability of the material comprising the core; in this context, we added PDMS-*b*-PEG to the set of block copolymers since PDMS has almost 2–3 orders of magnitude higher oxygen permeability [44] compared to other block copolymers of the series.

The choice of block lengths was the compromise of three requirements: (i) the hydrophobic block should be long enough (DP of ca. 50 or more) to provide low CMC values for high micellar stability towards de-aggregation [24,25]; (ii) *M_n_*(PEG)/*M_n_*(hydrophobic block) ≥ 1 to ensure spherical morphology of micelles since increase in hydrophobic block content leads to non-spherical morphologies [23] and the loss of colloidal stability of micelles [25]; and (iii) the chosen block DP values should be as close as possible to each other to exclude the influence of this parameter on the micellar properties. As a result, we have chosen six block copolymers with DP(PEG) = 110 ± 25 (*M_n_*(PEG) = 5000 ± 1200) and DP(hydrophobic block) = 45 ± 10 with two exceptions for hydrophobic block lengths: PBd_90_-*b*-PEG_130_ and PDMS_15_-*b*-PEG_115_. All the block copolymers used in the study demonstrated a reasonable agreement of their properties with those claimed by the manufacturers (Table 1).

### 3.2. Hydrodynamic and Molecular Characteristics of Block Copolymers in Organic Solvents

The molecular properties of block copolymers were studied in two organic solvents: THF (viscometry, densitometry, AUC, DLS, GPC) and DMSO-*d_6_* (^1^H NMR). THF was used as the main solvent due to the high solubility of all the block copolymers, and also due to the opportunity to compare our data with those from SLS experiments for similar block copolymers in the same solvent [17].

Figure 1 presents the normalized distributions (*c(s)*_norm_) of sedimentation coefficients *s* obtained by AUC implementing velocity sedimentation method and resolved using Sedfit program. All block copolymer samples were studied under the same conditions (THF solutions, 20 °C and maximum possible rotor speed of 60,000 rpm to ensure best resolution of sedimentation profiles). The obtained distributions (Figure 1) are relatively narrow with sedimentation coefficients in the range typical for low molecular weight macromolecules with dispersities close to 1, which correlates well with GPC data (Table 1; Appendix A).

Comparison of the obtained distributions leads to the following considerations. First, half of the studied samples (Figure 1B) demonstrate a minor peak within resolved distributions next to y-axis. Its nature has been discussed earlier [45], and the origin of this peak can be caused by either the presence of low molar mass impurities or biases in numerical resolution of the partial differential Lamm equation [46] incorporated in Sedfit program. Taking into account the above-considered factors and both negligible *s* values and areas under the distribution curves determined for the minor peaks within resolved distributions next to the y-axis, these peaks were ignored in further analysis. Second, the distributions obtained for the PS-*b*-PEG, PDMS-*b*-PEG and PCL-*b*-PEG are the narrowest ones, which is in a good correlation with GPC data. Indeed, all three samples are characterized by low dispersity values *Đ* ≤ 1.15 (Table 1). Third, the distributions acquired for PMMA-*b*-PEG, PE-*b*-PEG and PBd-*b*-PEG are the widest and the latter two demonstrate apparent “bimodality”, which is rather an artifact in Sedfit analysis common for the synthetic polymers with finite but continuous dispersity [35]. Besides, PMMA-*b*-PEG and PE-*b*-PEG samples have the widest *c(s)*_norm_ distributions and the highest dispersity values according to GPC (1.22 and 1.28, respectively; Table 1). Regarding PBd-*b*-PEG, the second (having higher *s*) mode could be also due to partial cross-linking of block copolymer chains featuring double C=C bonds in the main chain of PBd block [47]. In the case of PDMS-*b*-PEG and PCL-*b*-PEG, we also cannot completely exclude partial decomposition of PDMS and PCL blocks, correspondingly, since both blocks contain potentially hydrolysable Si-O bonds (PDMS) and ester groups (PCL) in the main chain. This undesirable process can be an alternative reason for slowly sedimenting species resulting in the appearance of low *s* value peaks in the case of these two block copolymers. In the further analysis, the weight average s values of the presented distributions (Figure 1) were used excluding low molecular peaks next to the y-axis. Finally, the aforementioned possible presence of unreacted macromonomer residues can be ruled out for PS-*b*-PEG and PMMA-*b*-PEG, since the corresponding distributions of *c(s)*_norm_ are unimodal (Figure 1A).

The concentration dependences of *η_sp_*/*c* were linear for all the studied samples (Figure 2). The values of intrinsic viscosity were obtained by Huggins procedure. It should be noted that the average value of Huggins constant *k_H_* equals to 0.3 ± 0.1, which indicates that THF may be considered as a thermodynamically good solvent for all the studied copolymers. The obtained values of intrinsic viscosity of the copolymers fall in the range from 0.13 to 0.25 dL/g (see Table 2), and this result is in good agreement with the intrinsic viscosity of PEO in THF [48].

The summary of hydrodynamic data of block copolymers in THF obtained by the combination of hydrodynamic methods is presented in Table 2. One can see that all the block copolymers have diffusion coefficients, hydrodynamic radii, intrinsic viscosities, and sedimentation coefficients characteristic for single macromolecules and agree well with the literature data, for example, PS_10_-*b*-PEG_70_ was reported [17] to have *R_g_* = 2.4 nm in THF. *M_w_* and dispersities (*Đ*) of block copolymers obtained for THF solutions by GPC are presented in Table 1. Some discrepancies (less than 1.5-fold) between *M*_sD_ calculated from the combination of DLS and AUC (Table 2) and *M_w_* obtained by GPC (Table 1) do not exceed analogous inconsistencies reported in the literature [17] and result from the fact that the presence of two chemically different blocks does not allow correct calculation of *M_w_* values based on homopolymers GPC standards. In contrary, *M*_sD_ values are absolute ones, and they are much closer to the true molar masses. Consequently, further estimations of micelle parameters were based on *M*_sD_ values. The above results point to the formation of molecularly dispersed solutions for all the block copolymers in THF and also in other solvents used in the study (DMSO, DMF).

The analysis of molecular hydrodynamic experiments (viscometry, velocity sedimentation (AUC), and diffusion (DLS)) makes it possible to characterize the macromolecules from the viewpoint of their rotation and translation mobility. One of the most important options of this approach is the opportunity to ensure the self-consistency of acquired hydrodynamic data based on the concept of hydrodynamic invariant. The values of hydrodynamic invariant *A_0_* were calculated by the formula $A_{0}={\left(k_{B}TN_{A}{\left[D\right]}^{2}\left[s\right]\left[\eta \right]\right)}^{1/3}$, where *k*_B_—Boltzmann constant; *N*_A_—Avogadro’s number; $\left[D\right]=D_{0}h_{0}/T$ and $\left[s\right]=s_{0}h_{0}/\left(1-\bar{v}\rho_{0}\right)$ are characteristic values of diffusion and sedimentation coeffcients, correspondingly. The obtained values of *A_0_* are presented in Table 2. The averaging of these magnitudes gives *A_0_* equal to (3.0 ± 0.4) × 10^−10^ g cm^2^ s^−2^ K^−1^ mol^−1/3^ that fits well with the data known for flexible uncharged linear macromolecules [49].

### 3.3. Preparation and Stability of Block Copolymer Micelles in Aqueous Dispersion

All the block copolymers appeared to be insoluble in water upon direct mixing and heating up to 60 °C, except for PDMS-*b*-PEG. The latter block copolymer was soluble in water under these conditions, but formed a rather turbid dispersion. Hence, we have chosen an alternative way of micelle preparation, the so-called *solvent exchange method* [1,2] followed by thorough dialysis. According to this method, water is slowly added into the starting solution of the block copolymer in a good organic water-miscible solvent at vigorous stirring [39]. Upon the addition of water, the solubility of the hydrophobic block in the solvent mixture steadily decreases, whereas the high solubility of the PEG block is retained. Under these conditions, nanophase separation occurs at a certain water content, leading to the formation of core–shell nanosegregated structures. At low water content, the cores are swollen by the organic solvent, and thus the micelles are equilibrium ones. Nevertheless, a further increase in water content (usually up to 60 to 80 vol. % of water) followed by dialysis eliminates the solvent from the cores. This method has several advantages in view of potential biological applications: the equilibrium nature of micelles at the early stages leads to the formation of small- and narrow-sized nanoparticles irrespective of the final state (equilibrium or ‘frozen’) of the micelles in the aqueous dispersion. As a result, the chosen method gives the lowest possible micelle sizes and the least possible secondary aggregation [18] while thorough dialysis at the final stage of micelle preparation effectively removes the traces of organic solvent to exclude the solvent-associated toxicity.

The choice of the starting organic solvent was determined by the requirements of high solubility of the block copolymer, full miscibility with water and as low as possible toxicity. The solvent that best meets all the above requirements is dimethyl sulfoxide (DMSO): it is the least toxic organic solvent fully miscible with water. Unexpectedly, all the block copolymers except for PDMS-*b*-PEG formed turbid water dispersions and partially precipitated after preparative centrifugation (15,000 rpm, 20,000× *g*, 15 min). On the contrary, PDMS-*b*-PEG formed almost transparent dispersions (much less turbid compared to direct dissolution in water) and yielded only traces of unstable fraction after centrifugation (Table 3). The reasons of inapplicability of DMSO for micelle preparation are unclear, but this result does not contradict the literature since we have not found any protocols that would use DMSO as a co-solvent for polymer micelle preparation. Using *N*,*N*-dimethylformamide (DMF; it is more toxic than DMSO but much less toxic than other organic solvents discussed below) yielded stable micelles for all block copolymers, except for PMMA-*b*-PEG and PE-*b*-PEG (Table 3). All these block copolymers (PS-*b*-PEG, PBd-*b*-PEG, PDMS-*b*-PEG, and PCL-*b*-PEG) formed almost transparent dispersions (Abs(500 nm) ≤ 0.005 for 0.5 mg/mL dispersions) that did not give any precipitate and demonstrated only a negligible decrease in their absorption spectra after preparative centrifugation (Abs(after centrif.)/Abs(before centrif.) ≥ 0.95; Appendix A). PMMA-*b*-PEG formed stable micelle dispersions from THF (Table 3). PE-*b*-PEG formed micelle dispersions with acceptable turbidity from 1,4-dioxane and *N*-methylpyrrolidone (NMP), and preliminary experiments revealed that PE-*b*-PEG demonstrated much narrower size and sedimentation coefficient distributions in the case of 1,4-dioxane. Nevertheless, these micellar dispersions yielded precipitates after preparative centrifugation, and hence this sample was “purified” by three cycles of micelle preparation from 1,4-dioxane followed by precipitate removal and freeze drying of the supernatant (the corresponding lyophilizate was the starting material for the next cycle). The resulting “purified” PE-*b*-PEG was used to prepare aqueous dispersions from 1,4-dioxane. Both PMMA-*b*-PEG micelles prepared from THF and PE-*b*-PEG micelles prepared from 1,4-dioxane revealed the same stability towards preparative centrifugation as all other block copolymers successfully prepared from DMF. Importantly, all the above micelle dispersions retained their stability after 2 months storage at 4 °C (Appendix A). Finally, we used THF to prepare PMMA-*b*-PEG micelles, 1,4-dioxane to prepare PE-*b*-PEG micelles while all other micelles were prepared from DMF.

Further assessment of micelles stability consisted in the estimation of their ability to reconstitute (i.e., to directly redisperse in water) after freeze-drying: such an ability is quite important in the context of practical applications since it makes it possible to store pre-formed micelles in dry state and prepare solutions of intended concentrations. Unfortunately, except for PDMS-*b*-PEG, all other micelles were unable to reconstitute after freeze-drying. This finding contradicts some reports that claimed successful redispersion of freeze-dried micelles [50] and puts an additional step of measuring micelle concentration (by weighting a freeze-dried aliquot, see the “Materials and Methods” section) for the correct determination of starting concentrations in stock solutions after the dialysis.

Finally, we estimated the apparent critical micelle concentration (CMC_app_) values for all stable dispersions by the pyrene solubilization method [24,25] (Table 3). CMC_app_ were obtained from concentration dependences of *I*_338_/*I*_334_ intensity ratios (Appendix A) calculated from excitation spectra of pyrene. For PS-*b*-PEG, PMMA-*b*-PEG, and PCL-*b*-PEG block copolymers, the CMC_app_ values were expectedly low (less than 2 mg/L), thereby pointing to the high stability of micelles towards disintegration. PS-*b*-PEG and PCL-*b*-PEG demonstrate good agreement with the literature data: for example, CMC_app_ = 3.2 mg/L [25] for PS_35_-*b*-PEG_235_ (in this case, slightly higher CMC_app_ is due to two times longer PEG block); CMC_app_ = 1.8 mg/L [27] for PCL_45_-*b*-PEG_115_ (our data (Table 3) equal to this value within the experimental error). For PBd-*b*-PEG and PE-*b*-PEG, CMC_app_ values are of the order of 10 mg/L. This result can be a consequence of low *T*_g_ (<RT) of the corresponding hydrophobic blocks and indicates increased mobility of unimer exchange (or, in other words, compromised stability of these micelles). Nevertheless, at least in the case of PBd-*b*-PEG, this shortcoming can be overcome by using post-preparation cross-linking of PBd core via double C=C bonds. For PDMS-*b*-PEG, CMC_app_ value is of 40 mg/L pointing to high lability of these micelles resulting from both low *T*_g_ and a very short PDMS block. It is thus not surprising that Ir(III) complex loaded into these micelles revealed high sensitivity to variations of composition of dispersion media [39]. As a consequence, practical applications of these micelles will require their additional stabilization, which is more challenging in this case because of the high inertness of both PDMS and PEG blocks.

In general, the combination of the data on the formation and stability of block copolymer micelles leads to the conclusion that in all the cases, except for PDMS-*b*-PEG, we deal with rather stable micelles that are expected to retain their integrity over time and will neither readily rearrange upon changing external conditions nor dissociate at high dilution. Consequently, the micelles investigated in this study seem to be very promising in various biomedical applications.

### 3.4. Hydrodynamic Behavior of Block Copolymer Micelles in Aqueous Dispersion

After the optimization of preparation protocols intended to obtain stable aqueous micellar dispersions, we investigated the micelles’ hydrodynamic properties by the combination of viscometry, DLS, SLS, and AUC. Aqueous micellar dispersions demonstrated rather complicated behavior: as a rule, the dispersions revealed non-unimodal distributions, and almost every system had its unique features. Below, we will first present a short summary of general similarities, and, second, each system will be discussed separately to stress its own features.

Table 4 presents summary of hydrodynamic parameters of block copolymer micelles obtained by the combination of DLS and SLS. Almost all the micellar dispersions (except those of PCL-*b*-PEG, *vide infra*) demonstrate bimodal particle distributions according to DLS data: small compact (*R_h_* ≤ 17 nm) particles and larger (50 < *R_h_* < 200 nm) ones. The smaller particles are typically interpreted in literature as spherical core-shell micelles [15,16,17]. One can see from Table 4 that all the micelles in the series demonstrate rather narrow variations in sizes (*R_h_* values vary from 10.5 to 17.0 nm), rather high *M_w_* (in order of 10^6^ g/mol) and rather low *A_2_* values (in order of 10^−5^ cm^3^mol/g^2^) typical for polymer micelles. The larger particles are typically interpreted in the literature as “loose micellar clusters” (“secondary micelle aggregates”) [15,16,17]; in most cases, their contribution into the scattered light intensity was very labile from batch to batch (Appendix A), but recalculation of their mass fractions revealed insignificant contribution of secondary aggregates (Appendix A).

Table 5 presents the summary of hydrodynamic parameters of block copolymer micelles obtained by the combination of DLS, viscometry, and AUC. AUC experiments demonstrated much more complicated and diverse distributions for micelles (Appendix A), but generally gave good agreement between different types of micellar masses (M*_sD_* and M_w_; M_w_ are 20 to 80% higher compared to M*_sD_*, most probably, due to different types of averaging; Table 4 and Table 5). It is worth noting that precise measurements of partial specific volume $\bar{\upsilon }$ values (needed for calculating the M*_sD_* values) were impossible by standard protocols of densitometry measurements and these values were obtained by using the *density variation approach* [34,35]. Measuring of *M*_sD_ values for both micelles (Table 5) and unimers (Table 2) allowed calculation of aggregation numbers, *N*_agg_. *N*_agg_ values (Table 5) ranged from ≤ 80 for block copolymers with the highest T_g_ of hydrophobic blocks (PS and PMMA), and *N*_agg_ ≈ 170 ± 50 for all other systems, except for PE-b-PEG micelles, which demonstrated exceptionally high M*_sD_* and *N*_agg_, vide infra. Finally, a combination of hydrodynamic data allowed calculations of *A**_0_* to give the average value equal to (2.4 ± 0.4) × 10^−10^ g cm^2^ s^−2^ K^−1^ mol^−1/3^ (Table 5). This result coincides well with experimental data obtained earlier for compact non-percolated macromolecules (2.7 × 10^−10^ g cm^2^ s^−2^ K^−1^ mol^−1/3^) within the experimental uncertainty [51,52,53,54].

The spherical symmetry of the investigated block copolymer micelles allows estimation of some structural parameters based on the corresponding *N*_agg_ and *R*_h_ values (Table 5). The calculations of core radii (*R*_core_), volume fractions (*φ*_core_), and thicknesses of corona (*R*_corona_) were performed assuming that (i) the micelles are spherically symmetrical and possess “core–shell” morphology; (ii) volume fractions of hydrophobic blocks in the cores equal to 1; and (iii) core densities were equal to those for bulk polymers. The resulting structural parameters of micelles are summarized in Table 6. The obtained data indicate that in all cases the core volume fraction does not exceed 16%, implying that all the micelles used in the present study can be described by the so-called “star” model featuring a compact spherical core and swollen corona [55]. In the case of PS-*b*-PEG, one can see that the structural parameters (*R*_core_ = 4.7 nm; *R*_corona_ = 9.0 nm) are in good agreement with similar estimations for analogous block copolymers (for example [18], for PS_38_-*b*-PEG_148_, *R*_core_ = 5.4 nm; *R*_corona_ = 6.0 nm). Taking into account that contour lengths for styrene and ethylene glycol monomer units are 0.341 [18] and 0.36 [15] nm, respectively, it is possible to estimate the overall contour lengths (L) of PS-*b*-PEG as follows: L(PS_35_) = 11.9 nm; L(PEG_115_) = 41.4 nm; L(PS_35_-*b*-PEG_115_) = 53.3 nm. It is clear that all the contour lengths are at least two times longer than the corresponding *R*_core_, *R*_corona_, and *R*_h_ values, i.e., both PS and PEG chains are not fully extended (though are substantially expanded). Similar conclusions can be made for other block copolymers, except PDMS-*b*-PEG, where the corresponding contour length of PDMS block (ca. 6 nm) only slightly exceeds *R*_core_ (4.4 nm). In this situation, the cores of the PDMS-*b*-PEG micelles should either consist of strongly stretched PDMS chains (that is quite unlikely due to high flexibility of PDMS chains) or include at least partially entrapped PEG chains, suggested earlier for PS_10_-*b*-PEG_X_ block copolymers (X = 10 and 20). In the latter case the micelles were reported to have increased *R_core_* values, most probably due to partial entrapment of copolymer chains into the PS core [18].

Below, we present short discussion of each particular system.

The bimodal distribution PS-*b*-PEG micelles revealed by DLS (Appendix A) is not unexpected since bimodality of PS-*b*-PEG dispersions was previously demonstrated for various block copolymers in both DLS [15,16,18] and AUC [17] experiments. The smaller particles (micelles) have typical *R_h_* values ranging from 8.6 nm (PS_10_-*b*-PEG_23_ [18]) to 23 nm (PS_113_-*b*-PEG_886_ [16]). For two examples of block copolymers (PS_38_-*b*-PEG_90_ and PS_38_-*b*-PEG_148_), close to our sample by their structural parameters, *R_h_* equal to 9.2 and 11.4 nm, while aggregation numbers, *N_agg_* were 110 and 104, respectively [18], the data obtained in the present study (Table 5) are in reasonable agreement with these values. The second mode appeared as a rather broad peak of larger particles (“loose micellar clusters”) with dimensions of ca. 50–200 nm somewhat larger than those described in literature (*R_h_* of ca. 40–70 nm) [15,16] and demonstrating much broader batch-to-batch variability (Appendix A) compared to micelles. Different estimations provide either low (ca. 1 wt.% or less) [25] or substantial (4 to 46 wt.%) [16] weight fraction of aggregates, depending on block copolymer structure and preparation protocol. The system under study (Appendix A) resembles the former case [25], most probably due to the finding that solvent exchange technique followed by dialysis suppresses aggregation more effectively compared to other preparation protocols [18]. In the velocity sedimentation experiments, polymer micelles feature two close narrow peaks, which are most probably Sedfit artifacts demonstrating finite micelles dispersity (Appendix A). The weight average sedimentation coefficient *s*_0_ obtained from sedimentation coefficient distribution (7.9 S; Appendix A) is reasonably higher than that of PS_10_-*b*-PEG_70_ micelles (ca. 4 S) [17], thus supporting the assumption that this mode corresponds to micelles.

The bimodal distribution for PMMA-*b*-PEG micellar dispersions (Appendix A) is less expected since bimodality of PMMA-*b*-PEG dispersions was not reported to date [32]. Interestingly, the PMMA-*b*-PEG micelles are the most compact micelles in the series; most probably, this is due to short PEG block (95 units vs. 115 or 130 units for other systems) resulting in thinner corona layer. The micelles revealed rather wide distribution in sedimentation experiments (Appendix A) featuring several narrow peaks in water and unimodal distribution in water/D_2_O mixture.

Similar to the PMMA-*b*-PEG dispersions, the bimodal distribution (revealed by DLS) for PBd-*b*-PEG dispersions was not reported to date (Appendix A) [12]. In sedimentation experiments, these micelles demonstrated unimodal distribution (Appendix A). Their sedimentation coefficient *s*_0_ in water was the lowest in the series (2.54 S; Appendix A); moreover, the micelles floated in water/D_2_O mixture. The both features may be ascribed to low density of PBd core, reflecting the highest value of the partial specific volume of the system within the studied series (Appendix A).

PE-*b*-PEG has the highest aggregation number, more than two times exceeding the corresponding *N*_agg_ values for the rest of block copolymers micelles. This can be explained by a very low solubility of PE in organic solvents: the signals of PE protons have not been found in the ^1^H NMR spectra in DMSO-*d_6_* (most probably, due to aggregation of PE blocks); in GPC experiments, PE-*b*-PEG reveals a high-molecular weight shoulder in GPC trace in THF (Appendix A), and stable aqueous dispersions were prepared only after the extensive PE-*b*-PEG purification. As a result, the preparation of the micelles can be compromised by incomplete PE dissolution in 1,4-dioxane, and the resulting non-equilibrium micelles. These speculations are corroborated to some extent by the observation that PE-*b*-PEG micelles have non-negligible fractions of micellar clusters (Appendix A). In sedimentation experiments, the micelles revealed unimodal distribution with a broad shoulder lasting up to 20 S, which might be associated with the micelles of bigger size (Appendix A).

PDMS-*b*-PEG micellar dispersions is the second system in the series featuring non-negligible fractions of micellar clusters (Appendix A). In sedimentation experiments, the micelles revealed unimodal distribution in both water and water/D_2_O mixture (Appendix A).

PCL-*b*-PEG micellar dispersions is the only system in the series that displayed an almost complete absence of micellar clusters (Appendix A). *R_h_* values obtained for PCL-*b*-PEG are in good agreement with those reported for PCL-*b*-PEG block copolymers having comparable block lengths: PCL_23_-*b*-PEG_45_ (12.5 nm; [26]) and PCL_45_-*b*-PEG_110_ (20 nm; [27]). In sedimentation experiments, the micelles showed wide distributions in both water and water/D_2_O mixture analogously to that of PMMA-*b*-PEG (Appendix A).

### 3.5. Cytotoxicity Study of Block Copolymer Micelles in Aqueous Dispersion

In the last part of the study, we evaluated the cytotoxicity of micelles by the standard MTT assay (Figure 3). We found that cell viability was more than 80% for all the concentrations investigated (up to 0.3 mg/mL for 24 h). This result demonstrates the high inertness of block copolymer micelles towards cells and thus their high potential in biomedical applications. Additionally, the MTT assay demonstrates the correct choice of the micelle preparation strategy since the residual amounts of organic solvents also do not induce any cytotoxicity.

## 4. Conclusions

In conclusion, in this report we have described a detailed investigation of hydrodynamic properties of a series of PEG-based amphiphilic diblock copolymers in both a molecular dispersed state in organic solvents and micellar aqueous dispersions. In the case of organic solvents, all the diblock copolymers form true solutions and their hydrodynamic behavior strongly resembles that of homopolymers in a good solvent. In the case of micellar dispersions, the majority of diblock copolymers gave only one dominating type of particles (“conventional” small and compact spherical micelles) while two systems (PE-*b*-PEG and PDMS-*b*-PEG) revealed non-negligible amounts of the second type of larger and heavier particles ascribed to micellar clusters. For both types of systems a complete hydrodynamic description allowed calculations of hydrodynamic invariants. In the case of organic solvents, *A*_0_ values are typical for flexible uncharged macromolecules, while for micellar aqueous dispersions, *A*_0_ is (2.4 ± 0.4) × 10^−10^ g cm^2^ s^−2^ K^−1^ mol^−1/3^, which is close to *A*_0_ values characteristic for spherical particles. To the best of our knowledge, the present study reports *A*_0_ values of block copolymer micelles in aqueous dispersion for the first time.

Additionally, in view of potential biomedical applications of the described block copolymer micelles, we have assessed their stability in dispersion (towards precipitation over time and at preparative centrifugation as well as towards disintegration upon dilution) and cytotoxicity. Our study shows that all the block copolymer micelles are non-toxic, and almost all of them (except for PDMS-*b*-PEG) form stable dispersions. Taking into account that the block copolymers used in the present study are commercial samples, one can conclude that these samples are suitable for numerous biomedical applications. It is worth noting that, though the studied copolymers are not cytotoxic by themselves, their potential biomedical applications might be limited by the type of the organic solvent used for the preparation of micelles, because this solvent should be common for both the block copolymer and the cargo. Nevertheless, variations in block copolymer chemistry and in the types of solvent provide a rather flexible platform for further optimization of preparation protocols for any particular application. In particular, the studied block copolymer micelles can be used as nanocontainers for phosphorescent organometallic complexes to build lifetime oxygen nanosensors as was proposed recently [39,43].

## Data Availability

Not applicable.

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
