# Peer review of "Amphiphilic Diblock Copolymers Bearing Poly(Ethylene Glycol) Block: Hydrodynamic Properties in Organic Solvents and Water Micellar Dispersions, Effect of Hydrophobic Block Chemistry on Dispersion Stability and Cytotoxicity"

_polymers, 2022, doi:10.3390/polym14204361_

Round 1

Reviewer 1 Report

The manuscript by Chelushkin, Tsvetkov et al. describes a systematic study on a series of PEG-based block copolymer micelles, which are relevant for biological applications. A detailed characterization of the polymers is provided, including the hydrodynamic behavior of these copolymers as single molecules, the preparation of micelles in water, their stability and hydrodynamic properties, and their cytotoxicity to model cells.

Although the paper does not reveal any particularly novel insight, it provides an important (and I believe unique) comparison between micelles that vary in the chemical compositions of their core. The considerations described in p. 6 attest to a rational and serious planning, and the paper is well arranged and written in an instructional way that is easy to read. Hence, I believe this study could be useful for researchers seeking to employ block copolymer micelles in their research after addressing the following concerns:

1.      The synthetic method by which the block copolymers were performed is not disclosed. This information is important, especially for the systems that showed bimodal size distributions. It is important to know which monomer was polymerized first, because dead polymer chains remaining from the first block synthesis may influence the micelle characteristics (especially if the dead chains are the hydrophobic block that forms the core of the micelle). Different synthetic approaches could also lead to different skeletal structures, which may strongly influence the results and their interpretation. For example, PE is depicted in Scheme 1 as a copolymer of ethylene and butylene (with an unknown composition). This hints to its formation by free radical polymerization. In this case, one would expect additional lengths of branches, which may strongly influence the glass transition temperature of these blocks. Alternatively, the PE block could have been synthesized using a Ziegler-Natta catalyst and a mixture of ethylene and butylene (to create a linear, non-crystalline PE). In this case the full characterization of this block should be provided.
The synthesis information should be obtained from the manufacturers (Polymer Source usually provides it in their catalogue) and the discussions in the text (p. 7) should be modified accordingly.

2.      Molecular characterization of the block copolymers is extensive, yet flawed. The authors chose to rely on the manufacturers’ data even when they disagreed with their experimental results (line 233). This is a questionable decision, because the largest difference was observed for the PCL-b-PEG polymer, which was the only polymer purchased from a different manufacturer than the other polymers. In this case, the experimental data obtained in the authors’ lab by the same procedure should actually allow a more accurate comparison between the systems than data provided from different sources. Additionally, while the Materials and Methods section describes how the experimental molar mass data was obtained (and unfortunately not used), it is unclear how the manufacturers analyzed their products. It should be noted that analyzing molar masses of block copolymers by GPC poses a real challenge, because standard calibration using homopolymer standards is bound to introduce considerable errors to the molar mass determination because of differences in hydrodynamic volumes of the different repeat units (both within the blocks and with that of the standards used for calibration). Hence, the molar masses of all polymers must be analyzed using a GPC equipped with an online viscoemeter and employing a universal calibration curve, which is the only reliable way to get meaningful molar masses of block copolymers by GPC. This is imperative for such a comparative study as the current one.

3.      The structural information on the micelle was obtained only indirectly, by analyzing data from scattering and sedimentation methods. Although the analysis was done meticulously, direct information from microscopy would complement the scattering data and considerably increase the scientific value of the paper. Hence, it is important to include cryo-TEM analyses of the micelles on the different systems and compare it to the scattering and sedimentation data.

4.      Discussion in page 12 on the determined sizes is lacking. The volume fractions of the core blocks is taken as 1 for all systems, in contradiction with the expected plasticization of the cores by residual water in the PCL and PMMA cases (as mentioned in lines 257-258). Additionally, the experimentally measured sizes are compared to the contour length; they should be related to the more physically meaningful end-to-end radii or the radii of gyration. Consideration of solvent quality and its influence on the hydrodynamic radii should also be included. I believe that with reliable molar mass data (see above), the consistency between theoretical prediction and experimentally measured sizes would be higher and provide more scientifically sound insights on the extent of stretching of the corona blocks and collapse (or extension due to residual solvent) in the core, which will make a more meaningful discussion.

Additional comments:

1.      Table S1 includes data that is central to the systems under investigation, thus it should be included in the main text, not in the Supporting Information. The standards used for determination of Mw should be mentioned in the caption.

2.      Line 263: “response onto” should be replaced with “respond to”

3.      Figure S1: GPC traces include illegible values on the axes, and axis titles are missing (especially the x-axis). This figure should be processed at a higher level for publication.

4.      Page 7: the discussion of the molar mass distributions of the systems includes confusing arguments and some unsubstantiated speculations. Explanations for the bimodality of some of the systems invoke large polydispersity as one of the factors, although the same argument applies to PMMA-b-PEG, which does not show bimodality (as the authors mention themselves). The relatively low molar mass of the systems is stated as another factor, without any explanation how it relates to bimodality. The arguments about the chemical stability of PCL and PDMS against hydrolysis and the tendency of PBd for cross-linking (during synthesis, I believe) are not substantiated by control experiments or literature references. Lastly, the option of remaining dead chains of the first polymerized block and other reasons that may originate from the synthesis are not discussed at all. This discussion should be revised after obtaining reliable information on molar masses and on the synthesis of each block copolymer.

5.      Line 456: the size range (9.8-17.0 nm) mentioned is inconsistent with the data provided in Table 3.

6.      Line 500: R(corona) = 8.6 is inconsistent with the data provided in Table 5.

7.      Line 504: L(PS35)=11.9 nm and L(PEG115)=41.4 nm are inconsistent with the contour lengths provided for the repeat units. It seems the contour length of the repeat unit of one polymer was used for the calculation of the contour length of the block of the other polymer by mistake.

8.      The cycotoxicity assay seems like an artificial addition, which provides no real insight on the difference between the micelles. The authors should consider adding another biological analysis that would address this difference to convert this part from a mere decoration to a meaningful addition that aligns with the main story.

Author Response

The response contains illustrations. The were not uploaded into this box and hense were attached as a file to this response.

Reviewer 1:

The manuscript by Chelushkin, Tsvetkov et al. describes a systematic study on a series of PEG-based block copolymer micelles, which are relevant for biological applications. A detailed characterization of the polymers is provided, including the hydrodynamic behavior of these copolymers as single molecules, the preparation of micelles in water, their stability and hydrodynamic properties, and their cytotoxicity to model cells.

Although the paper does not reveal any particularly novel insight, it provides an important (and I believe unique) comparison between micelles that vary in the chemical compositions of their core. The considerations described in p. 6 attest to a rational and serious planning, and the paper is well arranged and written in an instructional way that is easy to read. Hence, I believe this study could be useful for researchers seeking to employ block copolymer micelles in their research after addressing the following concerns:

  1. The synthetic method by which the block copolymers were performed is not disclosed. This information is important, especially for the systems that showed bimodal size distributions. It is important to know which monomer was polymerized first, because dead polymer chains remaining from the first block synthesis may influence the micelle characteristics (especially if the dead chains are the hydrophobic block that forms the core of the micelle). Different synthetic approaches could also lead to different skeletal structures, which may strongly influence the results and their interpretation. For example, PE is depicted in Scheme 1 as a copolymer of ethylene and butylene (with an unknown composition). This hints to its formation by free radical polymerization. In this case, one would expect additional lengths of branches, which may strongly influence the glass transition temperature of these blocks. Alternatively, the PE block could have been synthesized using a Ziegler-Natta catalyst and a mixture of ethylene and butylene (to create a linear, non-crystalline PE). In this case the full characterization of this block should be provided. The synthesis information should be obtained from the manufacturers (Polymer Source usually provides it in their catalogue) and the discussions in the text (p. 7) should be modified accordingly.

Response: In this entry, at least three issues can be addressed separately:

  1. The request for describing the synthetic methods used to prepare block copolymers. Indeed, the information on synthesis of block copolymers provided by “Polymer Source” is freely available and is more or less comprehensive; but it is not available for PCL-b-PEG manufactured by “Sigma-Aldrich”. Consequently, in Supporting Information file we have described the syntheses for five of block copolymers and provided the most reasonable speculations regarding the PCL-b-PEG synthesis (Supplementary Material file, Part 1, Page 2):

All block copolymers used in the study were commercial. According to the manufacturers, they were prepared as follows:

PS-b-PEG was prepared by living anionic polymerization, with PS block being prepared first and used as a macroinitiator for subsequent ethylene oxide polymerization. PMMA-b-PEG was prepared using OMe-PEG-OH as a macroinitiator for PMMA polymerization according to the following scheme:

PBd-b-PEG was prepared by living anionic polymerization, with hydrophobic blocks being prepared first followed by ethylene oxide polymerization. PBd block microstructure is 1,4-rich but contains products of 1,2-addition. PE-b-PEG was prepared by hydrogenation of PBd-b-PEG. PDMS-b-PEG was prepared by living anionic polymerization of hexamethyl cyclotrisiloxane followed by hydrosililation reaction with allyl PEG using Pr catalyst according to the following scheme:

The synthetic scheme for PCL-b-PEG was not disclosed, but we can speculate that the block copolymer was prepared by a ring-opening polymerization of caprolactone on OMe-PEG-OH macromonomer.

  1. The concerns on the presence of dead hydrophobic blocks in block copolymer samples. Indeed, the syntheses of PS-b-PEG, PBd-b-PEG, PE-b-PEG, and PDMS-b-PEG (see above) can contain dead hydrophobic chains: PS-b-PEG and PBd-b-PEG were polymerized starting from the respective hydrophobic block; PE-b-PEG was prepared from the corresponding PBd-b-PEG block copolymer by hydrogenation of PBd block, and thus also can contain hydrophobic PE homopolymer; PDMS-b-PEG was prepared via conjugation of end-functionalized PDMS and PEG homopolymers, therefore, presence of both blocks is possible. PMMA-b-PEG and PCL-b-PEG were prepared using OMe-PEG-OH as a macroinitiator, therefore only OMe-PEG-OH homopolymer can be present as an admixture.

As a result, possible presence of hydrophobic blocks can contribute into bimodality of the micelles. We have discussed this issue in the main text in the following way (Pages 5-6):

It is important to note that, according to preparation schemes provided by the manufacturer (see Part 1 of Supplementary Information for more detail), at least four block copolymers (PS-b-PEG, PBd-b-PEG, PE-b-PEG, and PDMS-b-PEG) can potentially contain hydrophobic homopolymer as an admixture. This possibility should be kept in mind in the discussion of hydrodynamic behavior of block copolymers in organic solvents and the secondary aggregation of block copolymer micelles in aqueous dispersion.

and on Page 8:

Finally, the aforementioned possible presence of unreacted macromonomer residues can be ruled out for PS-b-PEG and PMMA-b-PEG, since the corresponding distributions of c(s)norm are unimodal (Figure 1A).

  1. Skeletal structure of PE block. As we mentioned above, PE-b-PEG was prepared by hydrogenation of PBd-b-PEG. Since the starting block copolymer contains some portion of 1,2-addition products randomly distributed along the chain, the resulting PE block can contain some amount of ethyl groups as branches. Due to this reason, as well as due to the fact that hydrogenation is non-stereoselective, we can suggest that PE block is non-crystallizable.

  1. Molecular characterization of the block copolymers is extensive, yet flawed. The authors chose to rely on the manufacturers’ data even when they disagreed with their experimental results (line 233). This is a questionable decision, because the largest difference was observed for the PCL-b-PEG polymer, which was the only polymer purchased from a different manufacturer than the other polymers. In this case, the experimental data obtained in the authors’ lab by the same procedure should actually allow a more accurate comparison between the systems than data provided from different sources. Additionally, while the Materials and Methods section describes how the experimental molar mass data was obtained (and unfortunately not used), it is unclear how the manufacturers analyzed their products. It should be noted that analyzing molar masses of block copolymers by GPC poses a real challenge, because standard calibration using homopolymer standards is bound to introduce considerable errors to the molar mass determination because of differences in hydrodynamic volumes of the different repeat units (both within the blocks and with that of the standards used for calibration). Hence, the molar masses of all polymers must be analyzed using a GPC equipped with an online viscoemeter and employing a universal calibration curve, which is the only reliable way to get meaningful molar masses of block copolymers by GPC. This is imperative for such a comparative study as the current one.

Response: First of all, we would like to make a little correction: in this work, we used Mn values provided by the manufacturer for the only purpose, namely, to calculate the polymerization degree values depicted in Scheme 1. We believe that the values provided by the manufacturer are correct, since Mw and Mn values of the all the starting blocks were measured independently by GPC of a macromonomer and seem to be reliable; measurements of Mn values of the second blocks were based on 1H NMR.

Second, we fully agree with the Reviewer that calculations of molar masses of block copolymer by GPC is a challenge. Consequently, we obtained MsD values for block copolymers (MsD are absolute values, and, for the case of block copolymers, these values are more correct than Mw and Mn values obtained from GPC) and used them in all the rest of calculations. This fact was stressed in the manuscript at Pages 9-10:

Some discrepancies (less than 1.5-fold) between MsD calculated from the combination of DLS and AUC (Table 2) and Mw obtained by GPC (Table 1) do not exceed analogous inconsistencies reported in the literature [15] and result from the fact that presence of two chemically different blocks does not allow correct calculation of Mw values based on homopolymers GPC standards. In contrary, MsD are absolute ones, and they are much closer to the true molar masses. Consequently, further estimations of micelle parameters were based on MsD values.”, and also mentioned on Page 12:

Measuring of MsD values for both micelles (Table 4) and unimers (Table 1) allowed calculation of aggregation numbers, Nagg.

Finally, we have made an attempt to perform more reliable measuring of Mw and Mn values for block copolymers by GPC with more reliable detection. Since we do not have a viscometric detector suggested by the Reviewer, we tried to perform detection by a combination of refractometric and light scattering detector. In principle, such a combination provides an opportunity to measure the absolute Mw and Mn values. In practice, the interpretation of LS data appeared to be even more challenging than in the case of conventional GPC, since we have found unexpectedly low molar masses (PS-b-PEG) or highly asymmetric peaks (PBd-b-PEG). Obviously, molar masses calculated from GPC traces with LS detection appeared to be less reliable than MsD values obtained from hydrodynamics experiments. Consequently, we decided to retain MsD values as the main basis for all the structural calculations performed in the paper.

  1. The structural information on the micelle was obtained only indirectly, by analyzing data from scattering and sedimentation methods. Although the analysis was done meticulously, direct information from microscopy would complement the scattering data and considerably increase the scientific value of the paper. Hence, it is important to include cryo-TEM analyses of the micelles on the different systems and compare it to the scattering and sedimentation data.

Response: In the case of two block copolymers (PDMS-b-PEG and PCL-b-PEG) we have reported spherical morphology by a conventional TEM in our previous publication (Elistratova, A.A. et al., Eur. Polym. J. 2021, 159, 110761, doi:10.1016/j.eurpolymj.2021.110761). While preparing this paper, we tried to investigate the rest of the micelles by the same method, but were not able to avoid the strong tendency of micelles towards surface aggregation into larger 100-200 nm particles during the adsorption on the coating. This drawback could be overcome with the help of cryo-TEM, as suggested by the Reviewer, but we do not have a possibility to do this (rather sophisticated) experiment using our Core Facilities.

  1. Discussion in page 12 on the determined sizes is lacking. The volume fractions of the core blocks is taken as 1 for all systems, in contradiction with the expected plasticization of the cores by residual water in the PCL and PMMA cases (as mentioned in lines 257-258). Additionally, the experimentally measured sizes are compared to the contour length; they should be related to the more physically meaningful end-to-end radii or the radii of gyration. Consideration of solvent quality and its influence on the hydrodynamic radii should also be included. I believe that with reliable molar mass data (see above), the consistency between theoretical prediction and experimentally measured sizes would be higher and provide more scientifically sound insights on the extent of stretching of the corona blocks and collapse (or extension due to residual solvent) in the core, which will make a more meaningful discussion.

Response: First, our considerations were based on the fact that PMMA and PCL are sufficiently hydrophobic and do not swell strongly in water. On the other hand, the micelle cores are quite compact and small (compared to the overall micelle sizes), and even if we assume swelling up to 30%, this will result in the increase of Rcore to the value of about 10%, which is less than the uncertainty in determination of this value.

Second, we took contour lengths of blocks as limiting cases for our estimations; obviously, the shell thickness cannot exceed the contour length of PEG, etc. We agree with the Reviewer that end-to-end radii or the radii of gyration are more physically meaningful but their determination requires independent experiments (for example, small-angle scattering techniques) or calculations using some model considerations (and the choice of a particular model is also ambiguous).

Additional comments:

  1. Table S1 includes data that is central to the systems under investigation, thus it should be included in the main text, not in the Supporting Information. The standards used for determination of Mw should be mentioned in the caption.

Response: Done. Now the Table S1 appears as the Table 1. The numbering of the rest of the Tables in the main text has been rearranged (in Supplementary Material, the Table S1 was filled with the data on solvent properties, and the numbering of the rest of supplementary tables did not change). The characteristics of PS standards used in GPC were mentioned in Section 2.3.

  1. Line 263: “response onto” should be replaced with “respond to”

Response: Done. Also, the paper was re-checked for any other typos/mistakes.

  1. Figure S1: GPC traces include illegible values on the axes, and axis titles are missing (especially the x-axis). This figure should be processed at a higher level for publication.

Response: Done. GPC traces were replotted and the updated plots include easily readable axis titles and values.

  1. Page 7: the discussion of the molar mass distributions of the systems includes confusing arguments and some unsubstantiated speculations. Explanations for the bimodality of some of the systems invoke large polydispersity as one of the factors, although the same argument applies to PMMA-b-PEG, which does not show bimodality (as the authors mention themselves). The relatively low molar mass of the systems is stated as another factor, without any explanation how it relates to bimodality. The arguments about the chemical stability of PCL and PDMS against hydrolysis and the tendency of PBd for cross-linking (during synthesis, I believe) are not substantiated by control experiments or literature references. Lastly, the option of remaining dead chains of the first polymerized block and other reasons that may originate from the synthesis are not discussed at all. This discussion should be revised after obtaining reliable information on molar masses and on the synthesis of each block copolymer.

Response: Done. The discussion on Page 7 was substantially rewritten in a following way (now it appears on Page 8):

Figure 1 presents the normalized distributions (c(s)norm) of sedimentation coefficients s obtained by AUC implementing velocity sedimentation method and resolved using Sedfit program. All block copolymer samples were studied at the same conditions (THF solutions, 20 °C and maximum possible rotor speed of 60,000 rpm to ensure best resolution of sedimentation profiles). The obtained distributions (Figure 1) are relatively narrow with sedimentation coefficients in the range typical for low molecular weight macromolecules with polydispersity’s close to 1, which correlates well with GPC data (Table 1).

Comparison of the obtained distributions leads to the following considerations. First, half of the studied samples (Figure 1B) demonstrate a minor peak within resolved distributions next to y-axis. Its nature have been discussed earlier [42], and the origin of this peak can be caused by either the presence of low molar mass impurities or biases in numerical resolution of the partial differential Lamm equation [43] incorporated in Sedfit program. Taking into account considered above factors and negligible both s values and areas under the distribution curves determined for the minor peaks within resolved distributions next to y-axis, these peaks were ignored in further analysis. Second, the distributions obtained for the PS-b-PEG, PDMS-b-PEG and PCL-b-PEG are the narrowest ones, which is in a good correlation with GPC data. Indeed, the all three are characterized by low polydispersity values Đ ≤ 1.15. Third, the distributions acquired to PMMA-b-PEG, PE-b-PEG and PBd-b-PEG are the widest and the latter two demonstrate apparent ‘bimodality’, which is rather an artifact in Sedfit analysis common for the synthetic polymers with finite but continuous polydispersity [32]. Indeed, PMMA-b-PEG and PE-b-PEG samples have the highest polydispersity values according to GPC (1.22 and 1.28, respectively; Table 1).

and further within the same paragraph (Page 8):

This undesirable process can be the reason of slowly sedimenting species resulting in appearance of low s value peaks in the case of these two block copolymers. In further analysis the weight average s values of the presented distributions (Figure 1) were used excluding low molecular peaks next to y-axis. Finally, the aforementioned possible presence of unreacted macromonomer residues can be ruled out for PS-b-PEG and PMMA-b-PEG, since the corresponding distributions of c(s)norm are unimodal (Figure 1A).

  1. Line 456: the size range (9.8-17.0 nm) mentioned is inconsistent with the data provided in Table 3.

Response: Corrected. Now this text appears as follows: “(Rh values vary from 10.5 to 17.0 nm)

  1. Line 500: R(corona) = 8.6 is inconsistent with the data provided in Table 5.

Response: Corrected. Now this text appears as follows: “(Rcore = 4.7 nm; Rcorona = 9.0 nm)”

  1. Line 504: L(PS35)=11.9 nm and L(PEG115)=41.4 nm are inconsistent with the contour lengths provided for the repeat units. It seems the contour length of the repeat unit of one polymer was used for the calculation of the contour length of the block of the other polymer by mistake.

Response: Corrected. The mistake was in exchanging the corresponding contour lengths of the repeat units of both monomers. Now this text appears as follows: Taking into account that contour lengths for PS and EG monomer units are 0.341 [16] and 0.36 [13] nm, respectively, it is possible to estimate the overall contour lengths (L) of PS-b-PEG as follows: L(PS35) = 11.9 nm; L(PEG115) = 41.4 nm; L(PS35-b-PEG115) = 53.3 nm.

  1. The cycotoxicity assay seems like an artificial addition, which provides no real insight on the difference between the micelles. The authors should consider adding another biological analysis that would address this difference to convert this part from a mere decoration to a meaningful addition that aligns with the main story.

Response: While designing this work, we tried to achieve two goals: i) to provide an overall picture for the general applicability of block copolymer micelles decorated by PEG in biomedical applications. One can easily find numerous examples of good biocompatibility and bioinertness of PCL-b-PEG micelles, while much less (or even nothing) is known on similar properties for micelles based on PBd-b-PEG, PE-b-PEG, PDMS-b-PEG. Our work demonstrated that all these micelles are highly biocompatible; ii) to obtain the reliable experimental basis for our own works related to phosphorescence lifetime imaging. This work is in progress now, and it does demonstrate strong differences between the block copolymers, but this work requires loading of micelles by phosphorescent complexes, and is definitely far beyond the scope of the present work.

Finally, we would like to thank the Reviewer 1 for his keen and valuable comments that uncovered several important issues to be corrected. We hope we were able to provide an adequate response to them and were able to improve the paper by doing this.

Reviewer 2 Report

The manuscript by Tsvetkov et al. presents an elaborate investigation on hydrodynamic study on a number of commercial diblock copolymers. The characterizations have been done by GPC, DLS and the polymers have been tested by cell viability and cell toxicity. Although the there is no remarkable difference in any property among the chosen polymers, the authors have studied them thoroughly to bring conclusion on less toxicity for CHO-K1 cells.

Therefore, I suggest acceptance of this manuscript after some minor corrections of typo and addition on relevant citations of recent times (last ten years).

Author Response

Reviewer 2:

The manuscript by Tsvetkov et al. presents an elaborate investigation on hydrodynamic study on a number of commercial diblock copolymers. The characterizations have been done by GPC, DLS and the polymers have been tested by cell viability and cell toxicity. Although the there is no remarkable difference in any property among the chosen polymers, the authors have studied them thoroughly to bring conclusion on less toxicity for CHO-K1 cells.

We would like to thank the Reviewer 2 for his high evaluation of the present work. Our response in the following:

Therefore, I suggest acceptance of this manuscript after some minor corrections of typo and addition on relevant citations of recent times (last ten years).

Response: Done. 12 new references were added with 7 ones published after 2015. These new references comprise both relevant citations of recent literature (including those suggested by Reviewer 3) and those that were added in response to comments of Reviewer 1. Also, the paper was re-checked for any other typos/mistakes.

Reviewer 3 Report

The paper submitted by Elistratova et al. deals with the colooidal characterization of several micellar systems based on different types of amphiphilic block copolymer having PEG as hydrophilic block.

This is a very interesting paper, well written and shows a very good understanding of the micellar systems by the authors. However, some corrections are needed before its publication:

1. the introduction section must be completed with several new articles concerning the micellization of block copolymers in aqueous and non aqueous solvents. some suggestion might be: https://doi.org/10.1002/app.45313; 

2. why the authors chose to use the precipitation technique for the preparation of micelles instead of the direct dialysis? it is well known that dialysis starting from a common solvent it the best method to prepare micelles starting from copolymers with a relatively high molar amount of the hydrophobic block.

3. the authors must indicate the molar percentage of the hydrophobic and hydrophilic blocks, respectively. in this way, it will be possible to better compare the different samples. 

4. line 430. add a reference for the crosslinking of the PB block via C=C bonds

5. in their conclusion section, the authors must indicate that even the studied copolymers are not cytotoxic, their biomedical application might be limited by the type of the organic solvent used for the preparation of micelles. in fact, for the preparation of drug-loaded micelles, also the drug must be soluble in the same organic solvent as the copolymer. then, the precipitation method used by the authors for the preparation of micelles, might be not very suitable for the preparation of the drug-loaded micellar systems as it may led to drug precipitation before self-assembly of the copolymer.

Author Response

Reviewer 3:

The paper submitted by Elistratova et al. deals with the colooidal characterization of several micellar systems based on different types of amphiphilic block copolymer having PEG as hydrophilic block.

This is a very interesting paper, well written and shows a very good understanding of the micellar systems by the authors.

We would like to thank the Reviewer 3 for his high evaluation of the present work. Our response in the following:

However, some corrections are needed before its publication:

  1. the introduction section must be completed with several new articles concerning the micellization of block copolymers in aqueous and non aqueous solvents. some suggestion might be: https://doi.org/10.1002/app.45313;

Response: Done. 12 new references were added with 7 ones published after 2015. These new references comprise both relevant citations of recent literature (Ref. [3], [14] and [19] in Introduction; the paper suggested by the Reviewer appears as Ref. [19]) and those that were added is response to comments of Reviewer 1.

  1. why the authors chose to use the precipitation technique for the preparation of micelles instead of the direct dialysis? it is well known that dialysis starting from a common solvent it the best method to prepare micelles starting from copolymers with a relatively high molar amount of the hydrophobic block.

Response: While designing this work, we tried to obtain the reliable experimental basis for our own works related to phosphorescence lifetime imaging. This work is in progress now, and it requires loading of micelles by phosphorescent complexes. We have recently demonstrated (Zharskaia, N.A. et al., Biosensors 2022, 12, 695, doi:10.3390/bios12090695) that some organometallic complexes tend to precipitate from water-organic mixtures in the case of prolonged stirring or incubation with the mixtures enriched by organic solvent. As a result, the micelle preparation protocol is optimized to add excess (3 volumes) of water into starting organic solution in less than 15 minutes.

  1. the authors must indicate the molar percentage of the hydrophobic and hydrophilic blocks, respectively. in this way, it will be possible to better compare the different samples. 

Response: In fact, molar percentages of the hydrophobic and hydrophilic blocks are listed in Table 1 (formerly Table S1, shifted to the main text by the Reviewer 1 recommendation) as Experimental N(PEG)/N(block).

  1. line 430. add a reference for the crosslinking of the PB block via C=C bonds

Response: Done. The newly added reference appears as Ref. [45].

  1. in their conclusion section, the authors must indicate that even the studied copolymers are not cytotoxic, their biomedical application might be limited by the type of the organic solvent used for the preparation of micelles. in fact, for the preparation of drug-loaded micelles, also the drug must be soluble in the same organic solvent as the copolymer. then, the precipitation method used by the authors for the preparation of micelles, might be not very suitable for the preparation of the drug-loaded micellar systems as it may led to drug precipitation before self-assembly of the copolymer.

Response: Done. The newly added text appears as follows: “It is worth to note that, though the studied copolymers are not cytotoxic by themselves, their potential biomedical applications might be limited by the type of the organic solvent used for the preparation of micelles, because this solvent should be common for both the block copolymer and the cargo. Nevertheless, variation in block copolymer chemistry and type of solvent provide a rather flexible platform for further optimization of preparation protocols for any particular application.

Round 2

Reviewer 1 Report

The revised version addressed most of my comments satisfactorily.

Regarding comment 4, I believe that using any theoretical model to calculate radii of gyration would have provided a more insightful comparison than comparing to the contour length. Yet, this is a minor concern, and I leave it to the authors to decide whether to address it or not.